# Brief Weekly Magnetic Field Exposure Enhances Avian Oxidative Muscle Character During Embryonic Development

**DOI:** 10.3390/ijms26115423

**Published:** 2025-06-05

**Authors:** Jasmine Lye Yee Yap, Kwan Yu Wu, Yee Kit Tai, Charlene Hui Hua Fong, Neha Manazir, Anisha Praiselin Paul, Olivia Yeo, Alfredo Franco-Obregón

**Affiliations:** 1Department of Surgery, Yong Loo Lin School of Medicine, National University of Singapore, Singapore 119228, Singapore; surylyj@nus.edu.sg (J.L.Y.Y.); lesleywu@nus.edu.sg (K.Y.W.); surcfhh@nus.edu.sg (C.H.H.F.); e1088132@u.nus.edu (N.M.); anishapraiselin@gmail.com (A.P.P.); oliviayeoyiqi@gmail.com (O.Y.); 2BICEPS Lab (Biolonic Currents Electromagnetic Pulsing Systems), National University of Singapore, Singapore 117599, Singapore; 3Institute for Health Technology and Innovation (iHealthtech), National University of Singapore, Singapore 117599, Singapore; 4Department of Biomedical Engineering, College of Design and Engineering, National University of Singapore, Singapore 117583, Singapore; 5School of Life Sciences and Chemical Technology, Ngee Ann Polytechnic, Singapore 599489, Singapore; 6Department of Physiology, Yong Loo Lin School of Medicine, National University of Singapore, Singapore 117593, Singapore

**Keywords:** PGC-1α, PPAR-α, mitochondria, oxidative muscle, magnetic mitohormesis, mitochondriogenesis, SIRT1, myogenesis, metabolic dysfunction

## Abstract

Maternal metabolic dysfunction adversely influences embryonic muscle oxidative capacity and mitochondrial biogenesis, increasing the child’s long-term risks of developing obesity and metabolic syndrome in later life. This pilot study explored the mechanistic basis of embryonic muscle metabolic programming, employing non-invasive magnetic field exposures. Brief (10 min) exposure to low-energy (1.5 milliTesla at 50 Hertz) pulsing electromagnetic fields (PEMFs) has been shown in mammals to promote oxidative muscle development, associated with enhanced muscular mitochondriogenesis, augmented lipid metabolism, and attenuated inflammatory status. In this study, quail eggs were used as a model system to investigate the potential of analogous PEMF therapy to modulate embryonic muscle oxidative capacity independently of maternal influence. Quail eggs were administered five 10-min PEMF exposures to either upward-directed or downward-directed magnetic fields over 13 days. Embryos receiving magnetic treatment exhibited increased embryo weight, size, and survival compared to non-exposed controls. Upward exposure was associated with larger embryos, redder breast musculature, and upregulated levels of *PPAR-α* and PGC-1α, transcriptional regulators promoting oxidative muscle development, mitochondriogenesis, and angiogenesis, whereas downward exposure augmented collagen levels and reduced angiogenesis. Exposure to upward PEMFs may hence serve as a method to promote embryonic growth and oxidative muscle development and improve embryonic mortality.

## 1. Introduction

Skeletal muscle is the largest tissue mass in humans and is intimately linked to systemic metabolism and overall health. Muscle establishes systemic metabolism via a system of muscle–adipose paracrine crosstalk that is instigated by muscle mitochondrial activation. Muscle contraction stimulates the secretion of a host of anti-inflammatory, regenerative, metabolic, and adipose thermogenic factors (myokines or exerkines) and extracellular vesicles, collectively known as the muscle secretome, into the circulation for systemic action [1,2]. In this regard, weight bearing under the constant force of the Earth’s gravitational field establishes the basal metabolic rate but may be insufficient to offset caloric intake. Sedentary lifestyles, when accompanied by excessive caloric intake, are hence characterized by increased adiposity (caloric excess), metabolic disturbances, and systemic inflammation. In response to dynamic mechanical loading under the force of gravity, muscle mitochondrial respiration is further stimulated during exercise, and muscle–adipose crosstalk is activated to improve systemic metabolism. Moreover, muscles exhibit remarkable phenotypic plasticity, adapting to metabolic demands, such as dynamic changes in ATP/AMP ratio, lactate concentration, and oxygen availability. To fulfill the diverse physiological demands of the organism, skeletal muscle has evolved to exhibit appreciable metabolic plasticity [3]. Whereas glycolytic muscles are capable of generating large forces but fatigue rapidly [4], oxidative muscles generate less power that can be sustained for much longer. Oxidative muscles are hence predominantly responsible for maintaining posture under gravity and for establishing basal metabolism [5,6]. Muscle metabolic plasticity has been described down to the level of maternal–fetal interactions. Maternal obesity has been shown to reduce the oxidative capacity of fetal skeletal muscle, associated with reductions in mitochondrial content and efficiency [7]. Oxidative muscles are characterized by a higher abundance of mitochondria, greater myoglobin content, and denser microvasculature networks, giving them a distinct red appearance [8,9]. Oxidative muscles also preferentially rely on calorie-rich, yet slowly oxidized fatty acids as energy substrates for mitochondrial respiration, which establishes a favorable physiological environment for improved systemic insulin sensitivity and overall metabolic health [10].

Exercise-induced increases in muscle mitochondrial respiration activate transcriptional cascades governing oxidative metabolic adaptations, survival, and regeneration. The transcriptional coactivator peroxisome proliferator-activated receptor gamma coactivator-1 (PGC-1α) is largely responsible for the establishment of the oxidative muscle phenotype. On the one hand, the depletion of ATP during exercise increases the AMP/ATP ratio, which triggers AMP-activated protein kinase (AMPK) pathways to replenish the ATP stores [11]. On the other hand, the heightened NAD^+^ levels resulting from increased mitochondrial respiration during exercise activate deacetylase, Sirtuin 1 (SIRT1), to stimulate mitochondriogenesis [12,13]. Concomitantly, these two pathways converge on PGC-1α to drive the expression of oxidative metabolism genes [5,6]. Conversely, high-intensity anaerobic exercise promotes the glycolytic muscle phenotype through lactate-mediated signaling separately from aerobic mitochondrial respiration. The accumulated levels of lactate stabilize hypoxia-inducible factor 1α (HIF-1α) under low oxygen availability [14], leading to the transcriptional activation of glycolytic genes and angiogenesis [15,16], which enhances the anaerobic glycolytic capacity of muscles.

Pulsed electromagnetic fields (PEMFs) have been shown to promote myogenesis via the establishment of metabolic adaptations that include PGC-1α and SIRT1 activation, increased mitochondrial biogenesis, enhanced antioxidant defenses, a switch in mitochondrial energy substrate utilization towards fatty acids [17,18], and the promotion of muscle–organ paracrine crosstalk [19,20]. Notably, magnetic field directionality was shown to be a decisive factor in determining the mitohormetic efficacy of PEMF treatment [19]. This is not an unexpected finding, given that, when explicitly examined, magnetic field directionality is a common determinant of bioelectromagnetic field efficacy [21]. In healthy muscle cells, brief (10 min) exposure to low energy (1.5 mT (milliTesla)) PEMFs was shown to stimulate the proliferation and differentiation of muscle cells towards an oxidative phenotype, reduce basal apoptosis, and increase telomere length [17]. Moreover, weekly PEMF exposure was also shown to preferentially promote oxidative muscle development and enhance systemic fatty acid oxidation in mice [18]. Additionally, a human study demonstrated that 8 weeks of PEMF exposure improved body composition by increasing skeletal muscle mass while decreasing visceral and total body fat [22]. Given the parallels between PEMF exposure and aerobic exercise, magnetic field exposure may represent a feasible approach to promote oxidative muscle development and achieve metabolic balance.

This study investigated whether brief (10 min) PEMF exposures of quail eggs, amounting to five sessions over 13 days, could promote muscle oxidative development, embryo growth, and survival. This study provided initial proof-of-concept that PEMF paradigms may be used to promote avian embryonic oxidative muscle development aside from maternal influence.

## 2. Results

### 2.1. Upward-Directed PEMFs Yield the Greatest Response in Quail Embryo Development

Evidence for PEMFs promoting myogenesis has previously been demonstrated [17,23,24,25,26,27]. Moreover, electromagnetic field directionality has been shown to be of critical consideration [19]. In this study, we investigated the effect of brief electromagnetic stimulation on the development of quail embryos. To this end, fertilized quail eggs were exposed to either downward- or upward-directed PEMFs (1.5 mT, 10 min) on five separate days evenly distributed over a period of 13 days (Figure 1A) in two different arrangements of the eggs (Figure 1B). Exposure to either 1.5 mT downward- (blue dots) or 1.5 mT upward- (green dots) directed PEMFs significantly increased the wet weight of the embryo (~+20%) compared to control (red dots) in the “apex-up” arrangement (Figure 1C). On the other hand, there was no significant effect on embryo weight when the eggs were exposed in the “stacked” arrangement. Similarly, the body length of the embryos was significantly increased (~+15%) after either 1.5 mT downward- or upward-directed PEMFs in the “apex-up” arranged configuration of the eggs (Figure 1D). Notably, 1.5 mT upward-directed fields yielded slightly greater increases in body weight and length. We also investigated whether PEMF treatment could improve the viability of the quail embryos pre-hatching (Figure 1E). Both downward- and upward-directed PEMF exposure of eggs in an “apex-up” arrangement increased embryo viability compared to unexposed control eggs, albeit not significantly. Egg weight reduction was also measured over the intervention period. The weight of the quail eggs exposed to PEMF showed a smaller decrease compared to unexposed eggs, hypothetically resulting from more efficient substrate utilization of the embryo for purposes of tissue biosynthesis in a closed system (the egg). These results suggest that PEMFs can accelerate egg development and improve embryo viability, given appropriate magnetic field orientation and egg arrangement.

### 2.2. PEMF Exposure Promotes Oxidative Red Muscle

Oxidative muscle is “redder” than glycolytic muscle due to greater mitochondrial density, elevated levels of myoglobin, and being more enriched in microvasculature [8]. A muscle color analysis was next performed. PEMF-treated quail embryos were noticeably larger than the control embryos (Figure 2A). The breast regions of the embryos were photographed under controlled lighting conditions (Figure 2B), followed by the images being converted into grayscale for final quantification of color (Figure 2C). The pixel intensity was plotted into a heatmap (Figure 2D), and an average histogram was generated for all three conditions (Figure 2E). Embryos exposed to 1.5 mT upward-directed fields had a significantly redder breast musculature compared to 0 mT controls.

Oxidative muscle development was also ascertained at the level of gene expression. Transcript levels of the oxidative muscle-associated genes, peroxisome proliferator-activated receptor alpha (*PPAR-α*), and Sirtuin 1 (*SIRT1*) were measured in response to directional PEMF exposure from the musculature of embryos harvested from eggs in the “apex-up” arrangement. PPAR-α is a transcriptional regulator for mitochondrial adaptations involving lipid oxidation and energy homeostasis in skeletal muscle [28]. Endurance training in humans induces an increase in PPAR-α, enhancing the expression of mitochondria and myoglobin-enriched, and thus redder, oxidative muscle [29]. Upward-directed PEMFs resulted in a significant upregulation of *PPAR-α* gene expression compared to control (Figure 2F). The *PPAR-α* expression of the quail embryos exposed to downward-directed PEMFs, although showing a similar trend, failed to achieve statistical significance.

SIRT1 is a NAD^+^-dependent protein deacetylase that is involved in metabolism and aging [30,31]. Downward-directed PEMFs resulted in a significant increase in *SIRT1* transcripts compared to control, whereas upward-directed fields produced a modestly smaller increase that did not achieve statistical significance (Figure 2G). Transcripts for the collagen subtypes, *COL1A1* and *COL3A1*, were significantly upregulated in response to downward-directed magnetic fields but not upward-directed magnetic fields (Figure 2H,I). The PGC-1α-coactivated transcriptional cascade governs mitochondriogenesis, which is imperative for oxidative muscle development [8,32]. *PGC-1α* transcript levels were most elevated by downward PEMF exposure (Figure 2J). Vascular endothelial growth factor (VEGF) is necessary for the establishment of a microcirculatory network [33]. VEGFA protein levels were significantly downregulated upon exposure to downward-directed PEMFs (Figure 2K). The sum of the data thus suggests that upward-directed fields produce an overall more beneficial effect by promoting oxidative muscle development without inducing collagen or impeding angiogenesis.

## 3. Discussion

This study examined the capacity of PEMF treatment to promote embryonic development, particularly of oxidative muscle. An analogous PEMF stimulation paradigm was previously demonstrated to favor oxidative muscle development in isolated skeletal muscle cells [17,19], mice [18], and humans [19,34]. Brief (10 min) exposure of muscle to low-energy PEMFs was previously shown to stimulate muscle regeneration, associated with increases in mitochondrial biogenesis and antioxidative defenses, improved muscle inflammatory status, and cytokine (myokine) release [20]. This magnetic mitohormetic paradigm essentially recapitulates the effects of endurance training that exploits the adaptive benefits of exercise-induced oxidative stress over mitochondrial function to foster oxidative muscle development [6,8].

PPAR-α and PGC-1α are also involved in the fatty acid oxidation of skeletal muscle [35]. Due to the elevated respiratory capacity of oxidative muscle, fatty acids are the preferential energy substrate to support mitochondrial respiration, translating to improved systemic insulin sensitivity and reduced adipose-associated inflammation [10,36]. The present study showed that upward PEMF stimulation increased *PPAR-α* gene expression, which is imperative for oxidative muscle development [8,28]. Accordingly, exposure to upward-directed PEMFs significantly increased the redness of the quail embryo breast musculature.

Defining features of oxidative muscle are augmented mitochondrial respiration and elevated PGC-1α transcriptional activity [6,8]. Oxidative muscle development is modulated by calcium regulatory pathways governed by the canonical transient receptor potential channel 1 (TRPC1) [37,38,39]. TRPC1 serves as an integrator of diverse forms of biophysical stimuli of developmental consequence [40]. A TRPC1–mitochondrial signaling axis has been identified in skeletal muscle that can be activated by magnetic fields [17] and sustained mechanical loading [6,8,41] to promote oxidative muscle development [6,8]. Specifically, PEMF exposure and interaction with gravity stimulate TRPC1-mediated Ca^2+^ entry, which, in turn, trigger a PGC-1α-coactivated transcriptional cascade, governing mitochondriogenesis, angiogenesis, and oxidative muscle development [8,32]. PGC-1α transcriptional coactivation results in phenotypic commonalities between exercise and PEMF treatment [18].

PGC-1α is activated by its deacetylation by SIRT1, which, in turn, activates VEGF signaling [42,43]. Surprisingly, downward-directed PEMFs enhanced PGC-1α expression but reduced VEGF expression. This seemingly contradictory result might be due to the described interaction between SIRT1 and HIF-1α, a transcription factor that is activated by hypoxic conditions [30]. Previous studies have shown that the upregulation of SIRT1 leads to the deacetylation and inactivation of HIF-1α, which consequently attenuates VEGF expression, a downstream target of HIF-1α [30]. An increase in SIRT1 may thus serve to inhibit both HIF-1α and VEGF activity. Accordingly, acute exercise has been shown to induce the expression of both HIF-1α and VEGF, whereas their basal expression levels may become attenuated as a form of training adaptation [44,45]. As PEMF treatment recapitulates some aspects of endurance training [6], it is plausible that extended PEMF exposure diminished HIF-1α and VEGF expressions accordingly. Although both up- and down-field directionalities increased PGC-1α expression, downward-directed fields preferentially increased collagen expression. Increased muscle collagen may represent a predisposition for muscular fibrosis [46]. With relevance to the food industry, PGC-1α transcriptional activation is associated with better meat quality across species [47,48,49,50,51].

An interplay between egg orientation and field directionality was also demonstrated. The importance of field directionality was previously demonstrated in in vitro studies, where downward-directed PEMFs elicited the greatest myogenic and secretome responses compared to upward-directed PEMFs [19,52]. Sensitivity to magnetic field orientation is conferred by an interaction of TRPC1 with a cryptochrome (CRY2), a class of flavoprotein implicated in bird navigation to geomagnetic fields and circadian rhythm regulation of the cell cycle [19,52,53]. Notably, selectivity for downward magnetic fields is lost with the silencing of CRY2 expression and depletion of flavonoids [19,52]. Here, we demonstrated that when a monolayer of eggs was arranged vertically with their apex pointed upward, upward-directed PEMF exposure elicited the best developmental response with regard to embryo growth and the oxidative character of the breast musculature. Downward field exposure also promoted growth and embryo viability, but deficits in angiogenesis and enhanced collagen accretion were observed. The eggshell does not pose an impediment to magnetic field penetrance as determined by measurements of the magnetic field intensity above and below a layer stack of eggs. The loss of response to magnetic fields observed upon the stacking of the eggs most likely arose from the random orientation of the embryos during their exposure to vertically aligned magnetic field lines. In contrast, the quail embryos would assume a more standard alignment with reference to field line directionality in the more uniform planar apex up orientation. Based on the first principles, orthogonal alignment of the long axis of the embryo to the magnetic field lines would produce the greatest current induction [54]. Magnetic field directionality will hence influence muscle development and downstream systemic metabolism.

## 4. Materials and Methods

### 4.1. Quail Eggs and the Incubation Conditions

Japanese coturnix quail eggs were purchased from Uncle William Pte Ltd. (Bukit Timah, Singapore). The eggs were incubated at 38.5 °C with 70% humidity for two weeks using a Rcom MX-50 egg incubator (AUTOELEX Co., Ltd., Gyeongsangnam-do, Republic of Korea). The weights of the eggs were recorded daily. After two weeks, the embryos were harvested, and final weight measurements were taken. Photographs of the embryos’ chest area were captured under controlled illumination conditions before a small breast tissue was excised and snap-frozen with liquid nitrogen.

### 4.2. Pulsed Electromagnetic Fields (PEMF) Exposure

The use and characteristics of the PEMF device utilized in the present study were previously described [17]. Briefly, the PEMF device produces spatially homogenous, time-varying magnetic fields, consisting of barrages of 20 × 150 µs on and off pulses for 6 ms at a repetition frequency of 15 Hz. The magnetic flux density rose to a predetermined maximal level within ~50 µs (~17 T/s) when the driving field amplitude was 1.5 mT. The quail eggs were oriented with their apex pointing upwards in a single layer or laid horizontally and stacked into two layers. Both arrangements received either 0 mT or 1.5 mT with a downward- or upward-directed magnetic stimulation as previously described [19]. A total of 5 stimulations spanning across two weeks were given to the eggs, with at least one day gap in between the magnetic stimulations.

### 4.3. Breast Tissue Musculature and Color Analysis

Images of the quail embryos were analyzed using ImageJ (Version 1.54p, National Institute of Health, Bethesda, MD, USA). Red intensity within the pectoralis muscle region of interest (ROI) was quantified using the histogram function on ImageJ, which divides the red intensity spectrum into 256 intensity values. The frequency of pixels at each intensity level was determined, and peak counts were normalized to the ROI area to account for variations. The normalized data from individual samples were pooled to generate a consolidated table of peak frequencies and intensities for the red channel. The pooled data were visualized as a heatmap and a line graph to facilitate analysis of the data. To ensure objective analysis, the quantification of redness was performed on condition-blinded images using ImageJ.

### 4.4. Quantitative RT-PCR

Quantitative real-time polymerase chain reaction (RT-PCR) was carried out using the SYBR green-based detection workflow. Briefly, the frozen pectoralis major tissues were pulverized into a fine powder using a mortar and pestle under liquid nitrogen, followed by lysis using Buffer RLT Plus provided in the RNeasy Plus Mini Kit (Qiagen, Hilden, Germany). Total RNA was extracted according to the protocol provided in the RNeasy Plus Mini Kit (Qiagen, Hilden, Germany). RNA samples were quantified by NanoDrop One (Thermo Fisher Scientific, Waltham, MA, USA), and 1 µg of RNA was reverse transcribed to cDNA using the iScript cDNA Synthesis kit (Bio-Rad, Hercules, CA, USA). The quantification of gene transcript expression was performed using SsoAdvanced Universal SYBR (Bio-Rad, Hercules, CA, USA) on the CFX Touch Real-Time PCR Detection System (Bio-Rad, Hercules, CA, USA). Relative transcript expression was determined using the 2^−ΔΔCt^ method, with *β-actin* as the reference gene. The primers used were as follows: *PPAR-α*, F: GCT TGT GAA GGC TGT AAG GG, R: ACT TGG CCT TCT CAG ACC TC; *SIRT1*, F: TGA CAG AGC TTC ACA TGC AAG, R: ACA GCG TCA TAT CGT CCA GT; *COL1A1*, F: GCG ACT GTA CTA CTC ACC CG, R: TAT CGT TGT ACG TCA GCC CG; *COL3A1*, F: ATC CTC CCC AGC CCA TTA GT, R: GGC CTA TCA TTC CAG CAG GG; *β-ACTIN*, F: TGA CAG GAT GCA GAA GGA GA, R: ATG GTC CGG CTT CAT CAT AC.

### 4.5. Western Blot

Briefly, the frozen pectoralis major tissues were pulverized into a fine powder using a mortar and pestle under liquid nitrogen and then homogenized with a Dounce Homogenizer in ice-cold radioimmunoprecipitation assay (RIPA) buffer containing 150 mM NaCl, 1% Triton X-100, 0.5% sodium deoxycholate, 0.1% sodium dodecyl sulfate, 50 mM Tris (pH 8.0), and protease (Nacalai Tesque, Kyoto, Japan) and phosphatase (Roche, Basel, Switzerland) inhibitors. The protein concentration of the soluble fractions was determined using the Pierce BCA Protein Assay kit (Thermo Fisher Scientific, Waltham, MA, USA). A total protein of 20 µg was resolved using 10% denaturing polyacrylamide gel electrophoresis and transferred to an Immun-Blot PVDF membrane (Bio-Rad, Hercules, CA, USA). Proteins on PVDF membranes were blocked using 5% low-fat milk in TBST containing 0.1% Tween-20, followed by overnight incubation with primary antibodies in SuperBlock TBS (Thermo Fisher Scientific, Waltham, MA, USA) at 4 °C. The antibodies used are listed in Table 1.

HRP secondary anti-mouse or anti-rabbit antibodies were diluted (1:3000, Bio-Rad, Hercules, CA, USA) in 5% milk. The membranes were visualized using the Odyssey Fc Imaging System (LI-COR Biosciences, Lincoln, NE, USA) after incubation with the Clarity Western ECL Substrate (Bio-Rad, Hercules, CA, USA) or SuperSignal West Atto Ultimate Sensitivity Substrate (Thermo Fisher Scientific, Waltham, MA, USA).

### 4.6. Statistical Analysis

All statistics were carried out using GraphPad Prism (Version 10, Dotmatics, Boston, MA, USA) software. Unless otherwise stated, statistical analyses were performed using one-way analysis of variance (ANOVA) to compare the values between two or more groups, followed by Bonferroni’s multiple comparisons test.

## 5. Conclusions

The presented Magnetic Mitohormesis paradigm is a natural biophysical stimulus that has been proven to be safe and developmentally relevant for diverse cell types, tissue classes, animals, and humans, particularly with reference to skeletal muscle development. This study presents initial evidence that analogous low-energy pulsed electromagnetic fields can serve to metabolically program fetal muscle development independently of maternal influence. Given the non-invasive, drug-free, and brief nature of this biophysical stimulation paradigm, further investigations in mammals are warranted. The metabolic benefits of PEMF will likely extend beyond embryonic muscle development. Enhancing embryonic oxidative muscle development may have far-reaching implications, such as improving metabolic health to bolster disease resistance and reducing the reliance on prophylactic antibiotics, aligning with the accelerated demands for antibiotic-free farming. Additionally, optimized muscle energy metabolism could improve thermoregulatory resilience, addressing a critical growing concern in most poultry operations. Future research should evaluate the long-term effects on hatchability, immune function, and growth performance to validate PEMF’s potential as a sustainable agricultural intervention.

## Figures and Tables

**Figure 1 ijms-26-05423-f001:**
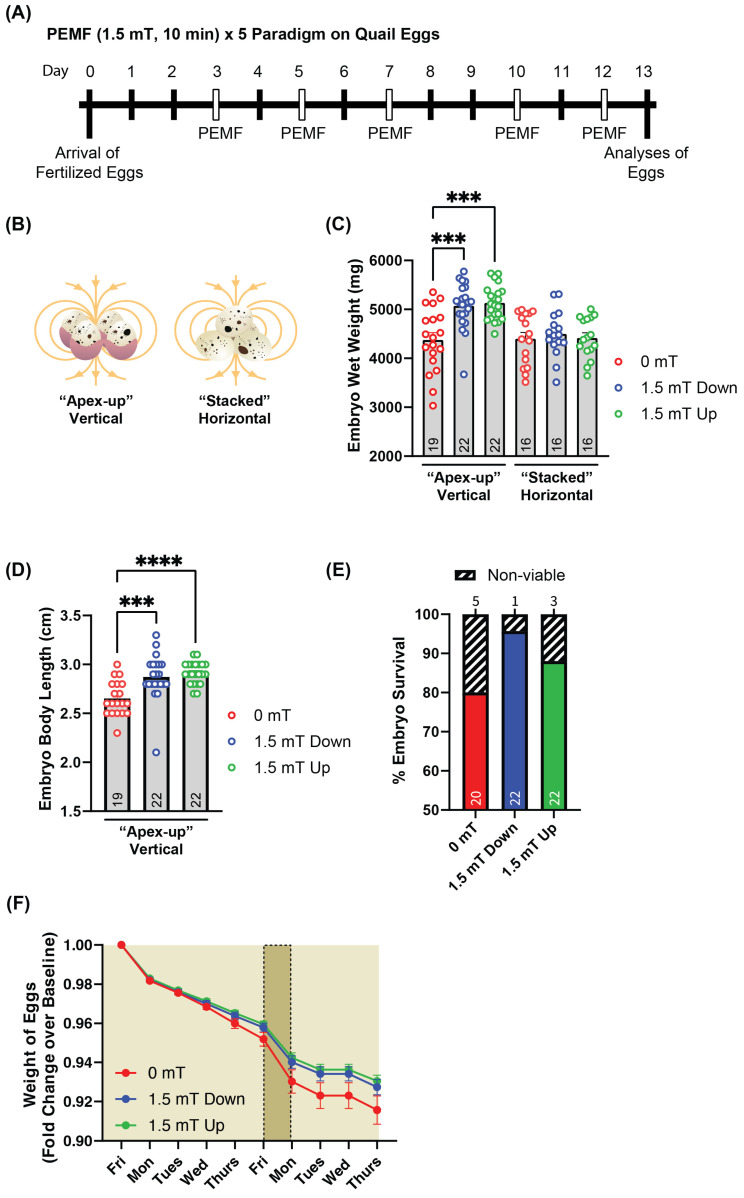
Effect of PEMF treatment on quail embryo development in ovo. (**A**) Fertilized quail eggs were exposed to either downward- or upward-directed PEMFs at an amplitude of 1.5 mT on five separate occasions, days 3, 5, 7, 10, and 12. Quail embryos were sacrificed for analyses on day 13. Egg weights were recorded daily. (**B**) Egg arrangements during exposure: “apex-up” vertically spread in a single layer or horizontally “stacked” in two layers. (**C**) Embryo wet weight after 13 days for quail eggs that were unexposed (control; red), exposed to 1.5 mT downward-directed magnetic fields (blue), or exposed to 1.5 mT upward-directed magnetic fields (green). PEMF exposure only elicited a significant effect on the wet weight of the embryos in the “apex-up” arrangement. (**D**) Embryo body length in the “apex-up” arrangement. (**E**) Embryo survival in percentage in the vertical “apex-up” arrangement. The absolute egg survival-to-dead ratios were 20:5 (0 mT), 22:1 (1.5 mT down), and 22:3 (1.5 mT up). (**F**) Egg weight throughout the entire incubation period is shown as fold change. Data represent mean ± SEM of embryo survival rates from two independent studies, with sample sizes (n = 16–22 eggs per condition) indicated within histogram bars. Statistical analysis was performed using the Kruskal–Wallis test, followed by Dunn’s multiple comparisons post hoc test. Significance is indicated by *** *p* < 0.0005 and **** *p* < 0.00005.

**Figure 2 ijms-26-05423-f002:**
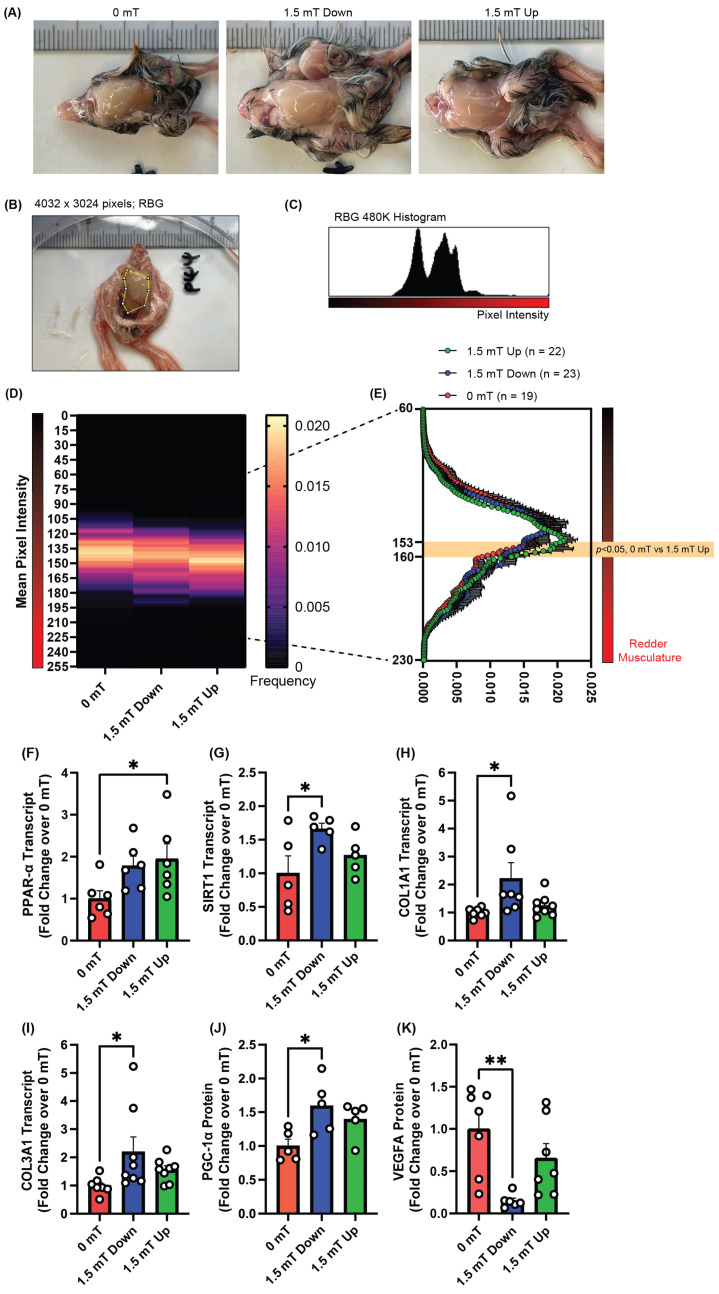
Effect of PEMF exposure on quail embryo chest musculature redness and modulation of gene and protein expression. (**A**) Representative embryo images for each experimental condition. Images were captured under controlled illumination conditions. (**B**) Representative region of interest of the pectoralis major, marked in yellow. (**C**) Representative 256-intensity histogram illustrating the pixel frequency and intensity distribution of an ROI in the red channel. (**D**) Heatmap showing the mean (consolidated) peak pixel frequencies and intensities of the red channel. The heatmap is organized according to intensity values (*y*-axis) with lower values representing less intense red pixels (top) and higher values representing more intense red pixels (bottom). (**E**) Line graph showing the peak means for intensities (*y*-axis) between 60 and 230. The error bars denote the standard error of the mean (SEM). Two-way ANOVA with Dunnett’s multiple comparisons post hoc test was used to compare the peak mean differences between the experimental groups within each intensity. (**F**–**I**) Fold changes in transcript levels of *PPAR-α*, *SIRT1*, *COL1A1*, and *COL3A1* detected by qPCR in quail breast tissue. Quail eggs were either unexposed (control; red), exposed to 1.5 mT downward-directed magnetic fields (blue), or 1.5 mT upward-directed magnetic fields (green); n = 5–8. (**J**,**K**) Fold changes in PGC-1α and VEGFA protein expressions. Data represent mean ± standard error of the mean (SEM). All statistical analysis was performed using one-way ANOVA, followed by a Bonferroni’s multiple comparisons post hoc test. Significance is indicated by * *p* < 0.05 and ** *p* < 0.005.

**Table 1 ijms-26-05423-t001:** List of primary antibodies used for Western blot analysis.

Antibody Name	Dilution Factor	Cat. No.	Manufacturer
PGC-1α	1:1000	66369-1-lg	Proteintech (Rosemont, IL, USA)
VEGFA	1:1000	PA1-16948	Thermo Fisher Scientific (Waltham, MA, USA)
GAPDH	1:10,000	60004-1-lg	Proteintech (Rosemont, IL, USA)
α-Tubulin	1:10,000	66031-1-lg	Proteintech (Rosemont, IL, USA)

## Data Availability

The original contributions presented in this study are included in the report. Further inquiries can be directed to the corresponding authors.

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
