# Peer review of "Brief Weekly Magnetic Field Exposure Enhances Avian Oxidative Muscle Character During Embryonic Development"

_ijms, 2025, doi:10.3390/ijms26115423_

Round 1

Reviewer 1 Report

Comments and Suggestions for Authors

Summary

This manuscript presents a proof-of-concept study that explores the effects of pulsed electromagnetic fields (PEMFs) on oxidative muscle development in avian embryos. The authors demonstrate that brief, non-invasive PEMF stimulation improves embryo growth, viability, and oxidative muscle characteristics, particularly when the field is applied in an upward direction. The study is timely and potentially impactful, especially given the increasing interest in non-pharmacological developmental interventions and the implications for both human health and poultry industry applications.

Major Strengths

  1. The use of PEMFs to modulate muscle development during embryogenesis—independent of maternal influence—is innovative. The concept of “magnetic mitohormesis” is well-articulated and supported by mechanistic links to known molecular pathways (e.g., PGC-1α, PPAR-α, SIRT1).
  2. The study integrates morphological (embryo weight and size), visual (muscle redness), molecular (gene and protein expression), and procedural (egg orientation, field direction) parameters to validate its conclusions.
  3. The PEMF device and stimulation parameters are clearly described, and statistical analysis appears appropriate for the exploratory nature of the work.

Major Concerns and Suggestions

  1. While the results clearly show a directionality-dependent effect of PEMFs, the mechanistic basis for this phenomenon remains speculative. The manuscript would benefit from inclusion (or at least discussion) of intracellular calcium flux measurements or TRPC1 activity assays to strengthen the link to the hypothesized Ca²⁺-PGC-1α axis.
  2. The significance of increased collagen (COL1A1, COL3A1) and decreased VEGFA in downward-directed PEMFs suggests potential fibrosis or impaired angiogenesis. However, no histological data are provided. Histopathological examination of muscle fiber types and vascular density would provide a much stronger validation.
  3. Since the effects were studied only up to day 13 of embryonic development, the long-term benefits (or risks) remain unknown. Including preliminary post-hatch follow-up data—even in a small cohort—would substantially enhance the translational value.
  4. Please describe the number of eggs used per group, especially for embryo survival in Figure 1E.

Minor Suggestions

- Clarify whether muscle redness quantification was blinded to treatment conditions.

- Consider replacing “bin” with other meaningful terms, such as “intensity range”.

- While directionality of PEMFs is introduced, the biological rationale for differential effects is not discussed until the results/discussion. A single sentence in Introduction hinting at this would improve logical flow.

- One of the authors holds a patent and is affiliated with a company commercializing PEMF technology. More explicit statements addressing how potential bias was mitigated would strengthen the integrity of the study.

Reviewer 2 Report

Comments and Suggestions for Authors

The manuscript by Franco-Obregón et al. explores the potential regulatory effects of pulsed electromagnetic fields (PEMFs) on embryonic muscle metabolic programming using a quail embryo model. The study is conceptually innovative, particularly in its effort to assess PEMF effects on embryonic development independently of maternal metabolic influence. The results suggest that upward-directed PEMF exposure may promote oxidative muscle development and mitochondrial biogenesis, offering a potential strategy for early intervention in metabolic diseases. However, there are several concerns regarding the experimental design, data presentation, and mechanistic interpretation. The manuscript requires additional data and discussion improvements before it can be considered for publication. Specific comments are as follows:

Although the study is presented as a pilot, it lacks clear information on the number of embryos per treatment group and the number of experimental replicates. This undermines the statistical credibility of the findings. The authors should specify group sizes and experimental replicates to strengthen the reliability of the conclusions.

The observed differences between upward and downward PEMF exposure are intriguing, but the underlying biophysical mechanisms are not well addressed. The authors should discuss possible mechanistic explanations with reference to existing literature. Additionally, they should consider whether the eggshell may attenuate or block magnetic field exposure, which could influence the interpretation of results.

Although upregulation of PPAR-α and PGC-1α transcripts is reported, no supporting data at the protein level (e.g., Western blotting or immunohistochemistry) or functional assays are provided. Transcript-level data alone are insufficient to confirm biological activity. The authors should include such analyses to validate the proposed mechanism.

The study focuses exclusively on embryonic development, without evaluating post-hatch metabolic or growth outcomes. A clearer discussion of the practical implications and limitations of the study is warranted, including potential long-term impacts of PEMF exposure.

In conclusion, while the study addresses a novel and potentially impactful topic, the current version lacks sufficient data and mechanistic depth. Based on this, I recommend minor revision and resubmission after the authors address the above concerns.

Comments on the Quality of English Language

The manuscript by Franco-Obregón et al. explores the potential regulatory effects of pulsed electromagnetic fields (PEMFs) on embryonic muscle metabolic programming using a quail embryo model. The study is conceptually innovative, particularly in its effort to assess PEMF effects on embryonic development independently of maternal metabolic influence. The results suggest that upward-directed PEMF exposure may promote oxidative muscle development and mitochondrial biogenesis, offering a potential strategy for early intervention in metabolic diseases. However, there are several concerns regarding the experimental design, data presentation, and mechanistic interpretation. The manuscript requires additional data and discussion improvements before it can be considered for publication. Specific comments are as follows:

Although the study is presented as a pilot, it lacks clear information on the number of embryos per treatment group and the number of experimental replicates. This undermines the statistical credibility of the findings. The authors should specify group sizes and experimental replicates to strengthen the reliability of the conclusions.

The observed differences between upward and downward PEMF exposure are intriguing, but the underlying biophysical mechanisms are not well addressed. The authors should discuss possible mechanistic explanations with reference to existing literature. Additionally, they should consider whether the eggshell may attenuate or block magnetic field exposure, which could influence the interpretation of results.

Although upregulation of PPAR-α and PGC-1α transcripts is reported, no supporting data at the protein level (e.g., Western blotting or immunohistochemistry) or functional assays are provided. Transcript-level data alone are insufficient to confirm biological activity. The authors should include such analyses to validate the proposed mechanism.

The study focuses exclusively on embryonic development, without evaluating post-hatch metabolic or growth outcomes. A clearer discussion of the practical implications and limitations of the study is warranted, including potential long-term impacts of PEMF exposure.

In conclusion, while the study addresses a novel and potentially impactful topic, the current version lacks sufficient data and mechanistic depth. Based on this, I recommend minor revision and resubmission after the authors address the above concerns.
